# Comparison of the Performance of Density Functional Methods for the Description of Spin States and Binding Energies of Porphyrins

**DOI:** 10.3390/molecules28083487

**Published:** 2023-04-15

**Authors:** Pierpaolo Morgante, Roberto Peverati

**Affiliations:** 1Department of Chemistry and Chemical Engineering, Florida Institute of Technology, 150 W. University Blvd., Melbourne, FL 32901, USA; 2Department of Chemistry, University at Buffalo, State University of New York, Buffalo, NY 14260, USA

**Keywords:** DFT, porphyrin, organometallic, density functionals, transition metals

## Abstract

This work analyzes the performance of 250 electronic structure theory methods (including 240 density functional approximations) for the description of spin states and the binding properties of iron, manganese, and cobalt porphyrins. The assessment employs the Por21 database of high-level computational data (CASPT2 reference energies taken from the literature). Results show that current approximations fail to achieve the “chemical accuracy” target of 1.0 kcal/mol by a long margin. The best-performing methods achieve a mean unsigned error (MUE) <15.0 kcal/mol, but the errors are at least twice as large for most methods. Semilocal functionals and global hybrid functionals with a low percentage of exact exchange are found to be the least problematic for spin states and binding energies, in agreement with the general knowledge in transition metal computational chemistry. Approximations with high percentages of exact exchange (including range-separated and double-hybrid functionals) can lead to catastrophic failures. More modern approximations usually perform better than older functionals. An accurate statistical analysis of the results also casts doubts on some of the reference energies calculated using multireference methods. Suggestions and general guidelines for users are provided in the conclusions. These results hopefully stimulate advances for both the wave function and the density functional side of electronic structure calculations.

## 1. Introduction

Porphyrins are a class of heterocyclic aromatic compounds that bind transition metals to form a broad family of coordination complexes, mostly known as metalloporphyrins. Due to their ubiquitous presence in biology and biochemistry—for example, in the active site of hemoglobin, myoglobin, and the cytochrome P450 family of enzymes—and their broad applicability as biomimetic catalysts, porphyrins have been extensively studied both experimentally and computationally [1,2,3,4,5,6,7,8,9,10,11]. The presence of the metal makes porphyrins challenging for electronic structure calculations due to several low-lying, nearly degenerate spin states [12,13,14,15,16,17,18]. Multireference treatments, such as those based on active-space methods such as CAS-SCF/CASPT2 (CAS = complete active space, SCF = self-consistent field, PT2 = Møller-Plesset perturbation theory truncated at second order), are usually necessary to correctly describe porphyrins and related compounds, as shown, for example, by Pierloot et al. [4,5]. Such calculations are not easily affordable due to their high computational cost and are usually limited to small systems. Variants such as the multiconfiguration pair-density functional theory (MC-PDFT) of Truhlar and Gagliardi et al. [19] appear promising for production calculations [9]. Single-reference methods based on Kohn–Sham (KS) density functional theory (DFT) account for static and dynamic correlation—at least in an approximate manner [20,21,22]—and thus represent a competitive alternative to multiconfigurational methods for large systems. In KS DFT, however, the accuracy of the approximation severely impacts the reliability of the calculated results [23,24,25,26,27], and choosing an appropriate functional might be a daunting task [28,29,30,31]. Benchmarking functional approximations is an effective way to understand the reliability of a functional against a wide range of chemical properties [23,24,25,26,27,32,33]. At the time of writing this article, no benchmark study aimed at assessing KS DFT calculations for metalloporphyrins is available in the literature. The main goal of this paper is to provide such a benchmark. This study reports calculations on spin state energy differences and binding properties of different metal porphyrins with 250 electronic structure methods, including 240 exchange-correlation functional approximations (a large majority of the functionals available in most electronic structure software). Suitable recommendations for choosing an appropriate electronic structure method for the computational study of porphyrins are provided at the end of Section 2.

## 2. Results and Discussion

### 2.1. Best Performers and General Trends

To concisely present the results for all 250 methods, we assigned grades to each functional based on their percentile ranking, as reported in Table 1. We note in passing that most of the grades obtained for the entire Por21 database are transferable to the PorSS11 and PorBE10 datasets (the individual rankings for the subsets are reported in the Appendix A). We set the threshold for a passing grade of D or better at the 60th percentile, corresponding to an MUE for Por21 of 23.0 kcal/mol. A total of 106 functionals achieved a passing grade, almost equally distributed among the grades A–D. Most of the grade-A functionals are local, either GGAs or meta-GGAs, with the addition of five global hybrids with a low percentage of exact exchange (r^2^SCANh, r^2^SCANh-D4, B98, APF(D), O3LYP). The GAM functional is the overall best performer for the Por21 database, ranking first for PorSS1 and second for PorBE10. Other notable grade-A functionals are all four parameterizations of HCTH and several revisions of SCAN, namely, rSCAN, r^2^SCAN, and r^2^SCANh. The three revisions perform better than the original functional, which has a grade of D. In particular, the r^2^SCANh and its -D4 variant stood out, with improvements larger than 50% over the errors obtained with SCAN. These results are consistent with other findings in the literature [34]. Three local Minnesota functionals are also in class A: revM06-L, M06-L, and MN15-L. These three functionals—together with r^2^SCANh and r^2^SCAN mentioned above—currently represent the best compromise between accuracy for general properties and accuracy for porphyrins chemistry and are at the top of our suggestion list (see below). For spin state energy differences, the results obtained in this work are consistent with the accepted knowledge that local functionals tend to stabilize low or intermediate spin states. In contrast, hybrid functionals stabilize higher spin states by including exact exchange [12,35].

Looking at the results for individual systems, we noticed that most functionals (233 out of 250) predict a triplet ground state for the iron porphyrin (FeP) system, while the CASPT2 reference predicts a quintet ground state (*vide infra*). Three top performers (GAM, HISS, and MN15-L) agree with the reference. The other results involving Fe(III) spin states also appear erratic. For the pentacoordinate FePOH system, 60% of the functionals agree with the CASPT2 references, predicting the high-spin state as the ground state. For the hexacoordinate FePNH_3_OH system, 72% of the functionals predict the ground state to be either the low- or intermediate-spin state, in contrast to the CASPT2 reference data. Finally, 90% of the methods predict the ground state of the d^7^ cobalt porphyrin to be a quartet, with spin state energy differences ranging from 1 to 90 kcal/mol (the CASPT2 reference is 0.90 kcal/mol in favor of the high-spin configuration). These results cast some doubts on the reference spin state energies for porphyrin containing Fe(III) and Co(II). 

For most functionals, the binding energies of the iron porphyrin with CO and NO appear unproblematic, regardless of the presence or absence of the imidazole ring. 90% of the functionals correctly predict the bound complexes to be more stable than the separate molecular fragments. The description of the binding with molecular oxygen appears to be more challenging: 60% of the methods predict the unbound FeP to be more stable than the FePO_2_ adduct, while only 25% of the methods predict the porphyrin-imidazole system to be more stable when unbound. This trend is observed for global hybrid functionals and range-separated functionals in both systems, and there is no apparent correlation between the stability of the adducts and the percentage of exact exchange employed in the functional form.

### 2.2. Results for Most Used and Most Suggested (MUMS) Functionals

To discuss the results in more detail, we selected a set of 25 functionals among the most used or the most suggested (MUMS) approximations for general and transition metal chemistry applications. The results are reported in Table 2. We note that selecting functionals is always subjective, but we tried to pick MUMS for their (unbiased) historical usage in the transition metal chemistry field [28,153,154] or following recent literature suggestions on methods that perform well for a broad range of properties [23,24,25,26,27,32]. When reporting standard functionals in the MUMS results, such as B3LYP or PBE, we included the parameterization without the dispersion corrections because we noticed that they generally worsen the MUEs for Por21. However, the magnitude of the worsening has no significant effect (usually within 5% or less) on the overall error of the method for all cases. As such, the results of standard functionals such as B3LYP and PBE can be transferred to the different dispersion correction ‘flavors’ (e.g., B3LYP-D3(BJ) or PBE-D2) without loss of generality. This consideration aligns with the fact that most interactions considered in Por21 are purely electronic. For example, dispersion corrections are almost entirely canceled out for the spin states when calculating the energy differences. For the bond energies, only a little overbinding residual remains, resulting in a modest worsening of the overall MUEs for methods that intrinsically overbind (which, in this case, is the vast majority). For functionals defined with dispersion corrections, such as PWPB95-D4 or ωB97M-V, we report their MUEs including such corrections, since this is how the functional was originally developed. Detailed results on each method can also be found in the Appendix A for more granular analysis.

The overall best performer among the MUMS approximations is the r^2^SCANh functional, with an MUE of 10.8 kcal/mol for the Por21 database, and 7.49 kcal/mol and 14.4 kcal/mol for the PorSS11 and PorBE10 subsets, respectively. The local Minnesota functionals M06-L and MN15-L come next, followed by r^2^SCAN, for a total of four functionals with MUEs < 15.0 kcal/mol (M06 is only 0.1 kcal/mol above this threshold). MN15-L is by far the best performer for the PorBE10 subset, with an MUE of 5.26 kcal/mol, but, unfortunately, it is only seventh for PorSS11, with an MUE of 17.9 kcal/mol. Looking at historically significant functionals, the PBE0 approximation performs well, followed by B3LYP. Modern transferable functionals such as B97M-V, PW6B95, SCAN, and MN15 position themselves in the middle of the pack, perhaps disappointingly, given their usually accurate performance for main-group elements [23,24,25]. TPSSh, traditionally regarded as one of the ‘gold standards’ of transition metal chemistry, ranks similarly with an MUE of 22.9 kcal/mol. On the disappointing side of the results, with MUEs higher than 25.0 kcal/mol, are some of the most popular semilocal functionals for transition metal applications, such as PBE, BP86, and TPSS. Range-separated hybrid functionals and double-hybrid functionals are even more disappointing, with MUEs higher than 30.0 kcal/mol. The latter class also presents challenges compared to most other functionals due to their computational cost and difficulty converging to the lowest energy solution. The notorious problems of the PT2-like correlation term for systems with multireference character easily explain these difficulties. For this reason, double-hybrid functionals are the only category where we did not explore every possible functional we had access to, but we limited ourselves to the most accurate approximations [142,155]. The B3LYP* is the worst performer among MUMS functionals for Por21, despite being specifically created to target spin state energy differences [70]. The improvements from B3LYP for PorSS11 are evident, but so are the deteriorating performances for PorBE10, where B3LYP is far superior. As already pointed out above, including dispersion corrections does not impact the results much, consistent with previous findings in the literature [35].

### 2.3. Results for Functionals Divided by “Ingredients”

An additional way to analyze the results is by classifying them according to how the approximation is constructed. For this purpose, we divided functionals into the following seven groups:**Group 0:** LDA, HF, LC, SE—local spin density approximations, Hartree-Fock, low-cost methods, and semiempirical methods (15 methods).**Group 1:** GGA—generalized gradient approximations and nonseparable gradient approximations (63 methods).**Group 2:** mGGA—meta-GGA and meta-NGA functionals (46 methods).**Group 3:** GH-GGA—global hybrid GGA and NGA functionals (43 methods).**Group 4:** GH-mGGA—global hybrid meta-GGA and meta-NGA functionals (43 methods).**Group 5:** RSH—range-separated hybrid functionals (28 methods).**Group 6:** DH—double-hybrid functionals (12 methods).

This classification follows well-established paths in the literature [23,28,156], and it provides valuable information to understand which ingredient is necessary for good performance. The distribution of the results for the methods in these groups is collected in the violin plots in Figure 1.

The plot analysis shows that the group of global hybrid GGA functionals (Group 3) is the best, with an average MUE of 19.8 kcal/mol. Groups 1, 2, and 4 perform similarly, with average MUEs of ~24 kcal/mol. The difference between Group 3 and Group 4 is particularly fascinating since global hybrid meta-GGA functionals, in principle, should be superior to global hybrid GGA functionals. To understand this (perhaps surprising) behavior, we report the MUEs of all functionals in both these groups against the percentage of exact exchange in each functional, as shown in Figure 2. From this plot, we notice a strong dependency of the MUE on the percentage of exact exchange, with the best results obtained by functionals with less than 30%, as expected [34,157,158]. As noted previously, B3LYP* is a particularly surprising outlier for the reasons we discussed in the previous section. The average percentage of exact exchange among the functionals in Group 3 is 23.0%, while the average percentage in Group 3 is 34.6%. This difference explains the superior performance of Group 3 functionals for the entire Por21 database compared to Group 4 functionals. If functionals with a percentage of exact exchange higher than 30% are removed from Group 4, then the average MUE for the remaining global hybrid meta-GGA functionals becomes identical to that of Group 3 (~20 kcal/mol). Additionally, we notice from the results for the subsets that the deterioration of the global hybrids meta-GGA results is mainly due to the PorSS11 database. The PorBE10 subset shows instead that Groups 2, 3, 4, and 5 have very similar average MUEs of ~22.0 kcal/mol, confirming that bond energies are far less sensitive to the choice of the method than spin state energy differences.

The disappointing performance of range-separated hybrid functionals in Group 5 is also easily rationalized based on a too-high percentage of exact exchange. This rationalization also applies to the double-hybrid functionals in Group 6. DH functionals usually include >50% exact exchange, which, together with the limitations already discussed in the previous section, are highly detrimental to their performance for Por21. Finally, Group 0 is perhaps an oddity since it includes methods with vastly different origins and ingredients. We grouped these methods because they do not belong to any of the other classes. Results reflect this non-uniformity, and given their poor performance, they will not be analyzed further.

When interpreting the results presented here, it is essential to remember that the MUE is a statistical parameter showing *average* performance. In other words, the MUE results of this study can be loosely interpreted as standard deviations (*σ*) and can be employed to give approximate error bars on metalloporphyrin calculations. In order to do so, however, we need to keep in mind that error bars of 1*σ* are not sufficiently stringent since individual errors might be (sometimes catastrophically) larger than the average one. This fact is confirmed by a granular analysis of our results, showing that the biggest individual error for one system can be as large as three times the average error. For example, our best performer, GAM, has an MUE of 6.93 kcal/mol, but its largest error is 17.7 kcal/mol. When using the stringent statistical threshold of 3*σ* (99.7 percentile of the error distribution), even the best performer gives error bars larger than 20 kcal/mol. If the less stringent 2*σ* threshold (95.5 percentile) is adopted instead, the error bars are smaller but still as significant as ~15 kcal/mol. For example, with a widely popular functional such as B3LYP, the 3*σ* and 2*σ* thresholds are ~60 and ~40 kcal/mol, respectively. Even for highly reliable functionals such as ωB97M-V, the 3*σ* error bars can be as large as ~100 kcal/mol, especially if the system is poorly described by a single determinant. This consideration is not intended as an endorsement in favor of B3LYP and against ωB97M-V but rather as a warning that the choice of the computational method should never be taken lightly [28,29,30,31], even if metals such as the closed-shell (d^10^) Zn(II) might seem less problematic at first glance. It is impossible to know a priori if the studied system requires a multireference treatment.

### 2.4. Discussion on Reference Energies and Chemical Accuracy for Transition Metals

The case of iron(II) porphyrin is of particular interest because—despite the ubiquity of this moiety in chemistry and biology—the energetic ordering of its spin states is still highly debated, even in recent literature [9,158,159]. The difficulties in describing FeP arise because the spin states are very close to each other, and even a small change in the geometry of the complex results in a reordering of the energy levels. This situation results in an intricate connection between the Fe–N distance and the predicted ground state, with the quintet state more stable at long distances and the triplet state more stable at short ones [159]. This complex dependence on geometry also complicates the interpretation of the experimental results since it is difficult to exclude the effect of the doming motion of the metal from the porphyrin plane in most experimental conditions. For completely co-planar iron and porphyrin—as in the Por21 geometry—CASPT2 predicts a quintet ground state, which is a reasonable prediction. However, given the complex nature of the energy states and considering the results of other reference calculations, the CASPT2 predicted energy difference of 7.00 kcal/mol between the quintet and the triplet seems suspiciously large. 

As emerges from the DFT results already discussed, single-reference methods should stabilize the low-spin states compared to high-spin ones. Pierloot and coworkers reported results from CCSD(T) calculations for FeP alongside the CASPT2 reference. These CCSD(T) results also present stabilization of the triplet, predicting a quintet ground state with a split of only 2.30 kcal/mol from the triplet, compared to the 7.00 kcal/mol predicted by CASPT2 (see Table 3). Most MUMS functionals predict a triplet ground state, except for MN15-L. On the one hand, we notice that the best functionals—r^2^SCANh and M06-L—are within 4 kcal/mol from the CCSD(T) results, although more than 8 kcal/mol from the reference CASPT2 results. On the other hand, MN15-L has an error of 5.6 kcal/mol compared to the CASPT2 reference but a difference of more than 10 kcal/mol from the CCSD(T) results. Similar trends, with significant differences between CCSD(T) and CASPT2 reference energies, appear also for the ^5^A_1g_ → ^3^E_g_ gap of FeP (see last column in Table 3) and the ^6^A_1g_ → ^4^A_2g_ and ^6^A_1g_ → ^2^E_g_ states of MnP. The ^5^A_1g_ → ^3^A_2g_ of FeP (not considered in Por21 for precisely this reason) and the ^4^B_1g_ → ^2^A_1g_ of CoP (see results in the Appendix A) appear less problematic.

Because of the complications highlighted above, these results are hard to interpret. It seems evident that the development of reliable, highly accurate reference electronic structure theory methods is required to address several of these issues. Lowering the cost of methods such as multireference coupled cluster [160], density matrix renormalization group (DMRG) [161], and quantum Monte Carlo (QMC) [162,163] might provide an answer in the future, even if those methods are not immune from interpretational difficulties (see Reference [164] for a recent unsolved controversy). Additional variants of the CAS methodology, such as the Stochastic Generalized Active Space (Stochastic-GAS) by Li Manni et al., also appear promising [158]. Given this situation, we still support the concept of “transition metal chemical accuracy” introduced by Wilson and coworkers [165]. Despite being proposed more than a decade ago, we reiterate the necessity to bring the chemical accuracy threshold to at least 3 kcal/mol for transition-metal-containing systems. The analysis of the results presented here and the available reference energies [9,11,158,159] suggest that a more conservative threshold could be considered at 5 kcal/mol for calculations on some of the most problematic metalloporphyrins. In the words of Wilson et al.: “This targeted accuracy is larger than for energetics of main group species because greater uncertainties are common in the experimental data for transition metal compounds and greater errors are expected with theory due to a number of factors, including increased valence electron space, stronger relativistic effects, and increased complexity of metal–ligand bonding.” [166].

### 2.5. Recommendations for Users and Final Remarks

In this work, we assessed the performance of 250 approximations for calculations of spin state energy differences and binding energies of Mn(II), Co(II), Fe(II), and Fe(III) porphyrins. Unfortunately, the average errors observed for these systems are very far from the chemical accuracy threshold of 1.0 kcal/mol that several functionals achieve for reaction energies of main-group elements [23,24]. These results reflect both an intrinsic difficulty of density functional calculations on metalloporphyrin and the difficulties in obtaining reliable reference energies and experimental results. We support raising such a chemical accuracy threshold to at least 3 kcal/mol for transition metals, as previously suggested by Wilson et al. [165,166].

In light of these underwhelming results, we believe that the purpose of this extensive benchmark should not be the identification of a single best performer but rather to suggest trends and guidelines. In this spirit, we provide the following general recommendations for calculations on porphyrin-containing systems:Avoid LDA, Hartree-Fock, and semiempirical methods (Group 0).Avoid functionals containing PT2-like correlation, such as double-hybrids (Group 6).Avoid range-separated hybrids (Group 5).Prefer semilocal functionals (Group 1 and Group 2) for spin states and other purely electronic properties.Among hybrid functionals in Group 3 and Group 4, prefer those with a percentage of exact exchange below 30%.For all other properties—including thermochemistry—prefer functionals that scored a grade of A or B in this study (Group 3 and Group 4) and are highly transferable to other systems across chemistry, as established in other benchmark studies. Some suitable suggestions are (our grades are in parenthesis, and high transferability is indicated in bold font):(1)Semilocal functionals: **MN15-L (A)**, GAM (A), revM06-L (A), M06-L (A), r^2^SCAN-D4 (A).(2)Global hybrids: **r^2^SCANh-D4 (A)**, M06 (B), PBE0-D3(BJ) (B).

The suggestions above are our (current) top recommendations for methods that retain a certain degree of transferability across a broad range of chemical properties.

## 3. Materials and Methods

### 3.1. Software and Settings

Most of the calculations in this study are single-point KS DFT calculations performed with the def2-TZVP Gaussian-type basis set [167] and the functionals listed in Table 1. Functionals belonging to all rungs of Perdew’s *Jacob’s ladder* of density functional approximations [168] were employed. The list of functionals includes almost all the approximations developed to date, spanning more than forty years of functional development. The effect of dispersion corrections was also studied by adding -D2 [36], -D3 (in different “flavors”) [24,37,38,39,40,41,42,43], or -D4- [44,45,46] corrections to some common functionals. Selected semiempirical and other low-cost methods were also included for comparison. The calculations with the low-cost methods employed the basis sets accompanying each method (MINIX [148] for HF-3c, def2-mSVP [146] for PBEh-3c, and mTZVP [89] for B97-3c). The majority of the calculations were performed using a development version of the Q-Chem quantum chemistry package [169]. The calculations with the APF(D) [47], B98 [103], HISS [55], and X3LYP [150] functionals and the semiempirical PM6 [54] and PM7 [56] methods were performed using the Gaussian16 program [170]. The xtb program [171] was used for calculations with the GFN1-xTB [135] and GFN2-xTB [136] semiempirical tight binding methods, while the standalone dftd4 program [172] was used to obtain the -D4 dispersion corrections for selected approximations. Among the available semiempirical and low-cost methods, we chose the most accurate according to recent benchmark studies [89,136,173]. A Lebedev grid with 99 radial and 590 angular points was employed to integrate all functionals, and the stabilities of the final solutions were checked to ensure the proper convergence of the electronic energies, allowing for symmetry-breaking when necessary. For double-hybrid functionals, the default frozen-core option for the PT2-like correlation was enabled for computational efficiency. 

### 3.2. The Por21 Database

The database used in this study is called Por21, as defined in the ACCDB collection [174]. The reference energies were obtained with the CASPT2 method by Pierloot, Radón, and coworkers [4,11]. The database includes 11 spin states [11] of Mn(II), Co(II), and differently substituted Fe(II) and Fe(III) porphyrins, and 10 binding energies [4] of the complexes between a model system of the heme group and three different diatomic molecules: NO, CO, and O_2_. To analyze the results in more detail, the spin states data have been grouped in a subset called PorSS11, while the bond energies data were grouped in the PorBE10 subset. All the molecular geometries are taken from the original publications [4,11], and were not re-optimized in this work. The geometry for iron porphyrin is optimized at the PBE0/def2-TZVP level [93,167], while all the remaining PorSS11 ones are optimized at the BP86/def2-TZVP level [62,75,167]. The spin multiplicities considered are sextuplet, quadruplet, and doublet for Mn(II) and Fe(III), quartet and doublet for Co(II), and quintet, triplet, and singlet for Fe(II). The binding energies are calculated using the following reaction: FeP(Im) + X → FeP(Im)X(1)
where FeP corresponds to the iron porphyrin, FePIm is the iron porphyrin bound to the imidazole ring, and X is either CO, NO, or O_2_. The geometries of the reactants in Equation (1) were optimized at the PBE0/def2-TZVP level, while the geometries of the products were optimized at the BP86/def2-TZVP level. The reader is referred to the original publications [4,11] for additional details. A representative example of the systems included in each subset is reported in Figure 3. The geometries used in the calculations are available on our group’s GitHub page [175].

## Figures and Tables

**Figure 1 molecules-28-03487-f001:**
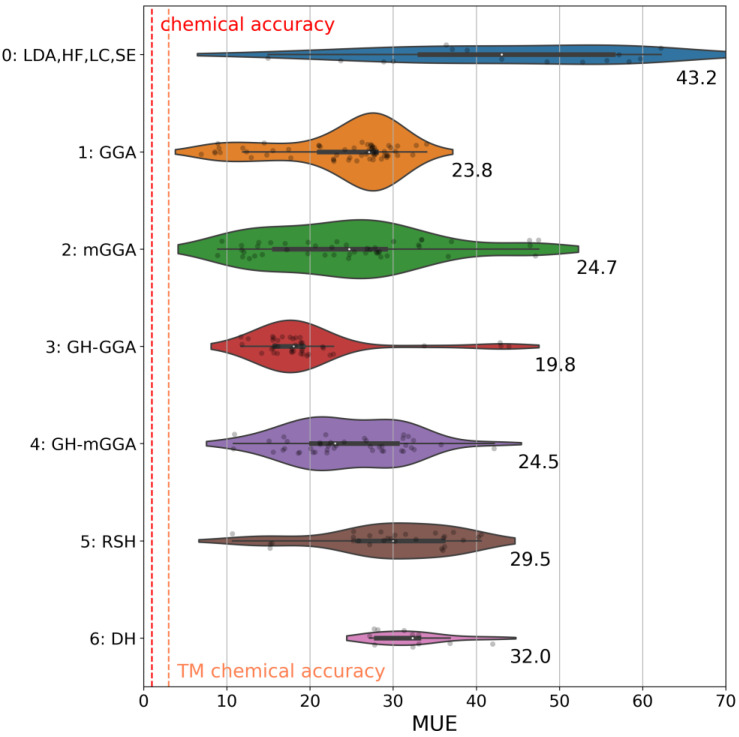
Distribution of the mean unsigned error (MUE) for the Por21 database for electronic structure methods divided into seven groups based on their ingredients (see text). Each group’s average MUEs is reported at the bottom-right of each violin. The area of each violin is proportional to the number of functionals in each group. The white dot in the center of the plot shows the median of the distribution. The thicker horizontal bar inside a violin shows the interquartile range of the data. The thinner black bar inside a violin shows 1.5 times the interquartile range of the data. Individual MUEs are also reported as black points within each violin (smoothness was applied; hence some outliers exist). Chemical accuracy and transition metal (TM) chemical accuracy (*vide infra*) are also reported (in red/orange, respectively, vertical dashed lines). All values in this plot are in kcal/mol.

**Figure 2 molecules-28-03487-f002:**
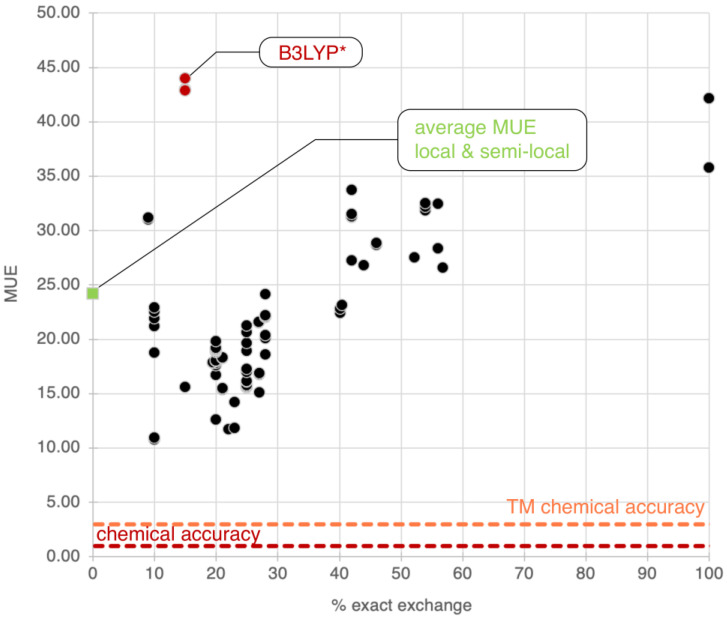
Distribution of the mean unsigned error (MUE) for the Por21 database for global hybrid functionals as a function of the percentage of exact exchange in each functional. The average MUE for the 107 local and semilocal functionals is also reported as a green square at 0%. The B3LYP* family is singled out as unusual outliners in red. Chemical accuracy and transition metal (TM) chemical accuracy are also reported. All values in this plot are in kcal/mol.

**Figure 3 molecules-28-03487-f003:**
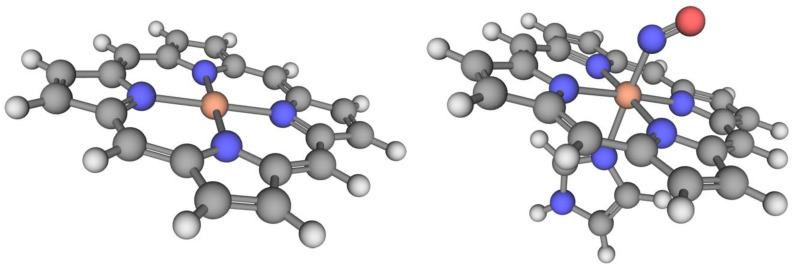
(**Left**) One of the iron-porphyrin rings (FeP) in the PorSS11 dataset. (**Right**) The porphyrin-imidazole-NO complex (FePIm-NO) from the PorBE10 dataset. The imidazole ring is used in place of the full heme active site. Hydrogen atoms are white, carbon atoms are black, nitrogen atoms are blue, the oxygen atom is red, and the iron atom is orange.

**Table 1 molecules-28-03487-t001:** List of methods examined in this study and their overall grade based on the MUE for Por21. Functionals are listed in alphabetical order, and grades are assigned based on percentile ranking, corresponding to the following thresholds: A: MUE < 14.3 kcal/mol; B: MUE < 17.1 kcal/mol; C: MUE < 20.0 kcal/mol; D: MUE < 23.0 kcal/mol; F: MUE > 23.0 kcal/mol. The dispersion corrections are covered in [24,36,37,38,39,40,41,42,43,44,45,46], unless noted otherwise. The MUEs were calculated from the CASPT2 reference energies; see the ‘Material and Methods’ section for further details. See also the Appendix A for results on the PorSS11 and PorBE10 subsets.

Functional	Grade	Functional	Grade	Functional	Grade
APF [47]	A	HFLYP [48,49,50,51]	F	PKZB [52]	D
APFD [47]	A	HFPW92 [48,49,50,53]	F	PM6 [54]	F
B2PLYP	F	HISS [55]	A	PM7 [56]	F
B2PLYP-D3(0)	F	HSE-HJS [57,58,59]	B	PW6B95 [60]	D
B2PLYP-D3(BJ)	F	HSE-HJS-D3(0)	B	PW6B95-D2 [61]	C
B2PLYP-D4	F	HSE-HJS-D3(BJ)	B	PW6B95-D3(0)	D
B3LYP [51,62,63]	C	LC-ωPBE08 [64]	F	PW6B95-D3(BJ)	D
B3LYP-D2	C	LC-ωPBE08-D3(0)	F	PW6B95-D3(CSO)	F
B3LYP-D3(0)	C	LC-ωPBE08-D3(BJ)	F	PW91 [65]	F
B3LYP-D3(BJ)	C	LC-ωPBE08-D3M(BJ)	F	PWB6K [60]	F
B3LYP-D3(CSO)	C	LRC-ωPBE [66]	F	PWB6K-D3(0)	F
B3LYP-D3M(BJ)	C	LRC-ωPBEh [66]	F	PWB6K-D3(BJ)	F
B3LYP-D4	C	M05 [67]	D	PWPB95-D3(BJ)	F
B3LYP-NL [68]	C	M05-2X [69]	F	PWPB95-D4	F
B3LYP* [70]	F	M05-2X-D3(0)	F	r++SCAN [71]	B
B3LYP*-D3(0)	F	M05-D3(0)	D	r^2^SCAN [72]	A
B3LYP*-D3(BJ)	F	M06 [73]	B	r^2^SCAN-D4 [74]	A
B3P86 [62,63,75]	C	M06-2X [73]	F	r^2^SCAN0 [76]	C
B3PW91 [62,63,65]	C	M06-2X-D2	F	r^2^SCAN0-D4 [76]	B
B3PW91-D2	C	M06-2X-D3(0)	F	r^2^SCANh [76]	A
B3PW91-D3(0)	C	M06-D2	B	r^2^SCANh-D4 [76]	A
B3PW91-D3(BJ)	C	M06-D3(0)	B	r^4^SCAN [71]	C
B97 [77]	C	M06-HF [78]	F	regTM [79]	F
B97-1 [80]	C	M06-HF-D3(0)	F	revM06 [81]	F
B97-1-D2 [61]	C	M06-L [82]	A	revM06-L [83]	A
B97-2 [84]	B	M06-L-D2 [61]	A	revM11 [85]	F
B97-2-D2 [61]	B	M06-L-D3(0)	A	revPBE [86]	D
B97-3 [87]	D	M08-HX [88]	F	revPBE-D2	D
B97-3-D2 [61]	D	M08-SO [88]	F	revPBE-D3(0)	D
B97-3c [89]	B	M11 [90]	F	revPBE-D3(BJ)	F
B97-D [91]	A	M11-D3(BJ)	F	revPBE-NL [68]	F
B97-D2 [36]	C	M11-L [92]	F	revPBE0 [57,86,93]	B
B97-D3(0)	C	M11-L-D3(0)	F	revPBE0-D3(0)	B
B97-D3(BJ)	C	mBEEF [94,95]	D	revPBE0-D3(BJ)	B
B97-K [96]	F	MN12-L [97]	F	revPBE0-NL [68]	C
B97M-rV [98]	C	MN12-L-D3(BJ)	F	revTPSS [99]	F
B97M-V [100]	C	MN12-SX [101]	F	revTPSSh [102]	C
B98 [103]	A	MN12-SX-D3(BJ)	F	RPBE [104]	D
BLOC [105]	F	MN15 [25]	F	RPBE-D3(0)	D
BLOC-D3(0) [104]	F	MN15-L [106]	A	RPBE-D3(BJ)	F
BLYP [51,62]	F	mPW91 [65,107]	F	rPW86PBE [57,108]	F
BLYP-D2	F	MS0 [109]	F	rPW86PBE-D3(0)	F
BLYP-D3(0)	F	MS0-D3(0) [110]	F	rPW86PBE-D3(BJ)	F
BLYP-D3(BJ)	F	MS1 [110]	F	rregTM [111]	F
BLYP-D3(CSO)	F	MS1-D3(0) [110]	F	rSCAN [71,112]	A
BLYP-D3M(BJ)	F	MS2 [110]	F	rSCAN-D4 [74]	B
BLYP-D4	F	MS2-D3(0) [110]	F	rVV10 [113]	F
BLYP-NL [68]	F	MS2h [110]	F	SCAN [114]	D
BMK [96]	F	MS2h-D3(0) [110]	F	SCAN-D3(0) [115]	F
BMK-D2 [61]	F	mTASK [116]	F	SCAN-D3(BJ) [115]	D
BMK-D3(0)	F	MVS [117]	A	SCAN-rVV10 [118]	F
BMK-D3(BJ)	F	MVSh [117]	D	SCAN0 [119]	D
BOP [62,120]	F	N12 [121]	F	SOGGA [57,122]	F
BOP-D3(0)	F	N12-D3(0)	F	SOGGA11-X [123]	D
BOP-D3(BJ)	F	N12-SX [101]	F	SOGGA11-X-D3(BJ)	D
BP86 [62,75]	F	N12-SX-D3(BJ)	F	SPW92 [53,124]	F
BP86-D2	F	O3LYP [125]	A	SVWN5 [124,126]	F
BP86-D3(0)	F	OLYP [51,125]	A	τ-HCTH [127]	A
BP86-D3(BJ)	F	OLYP-D3(0)	B	τ-HCTHh [127]	B
BP86-D3(CSO)	F	OLYP-D3(BJ)	B	TASK [128]	B
BP86-D3M(BJ)	F	OPBE [57,125]	B	TM [129]	F
BPBE [57,62]	D	oTPSS-D3(0) [37]	F	TPSS [130]	F
BPBE-D3(0)	F	oTPSS-D3(BJ) [37]	F	TPSS-D2	F
BPBE-D3(BJ)	F	PBE [57]	F	TPSS-D3(0)	F
CAM-B3LYP [131]	F	PBE-D2	F	TPSS-D3(BJ)	F
CAM-B3LYP-D3(0)	F	PBE-D3(0)	F	TPSS-D3(CSO)	F
CAM-B3LYP-D3(BJ)	F	PBE-D3(BJ)	F	TPSSh [132]	D
DSD-PBEP86-D3(BJ) [133]	F	PBE-D3(CSO)	F	TPSSh-D2 [61]	D
DSD-PBEPBE-D3(BJ) [133]	F	PBE-D3M(BJ)	F	TPSSh-D3(0)	D
GAM [134]	A	PBE-D4	F	TPSSh-D3(BJ)	D
GFN1-xTB [135]	F	PBE0 [93]	B	TPSSh-D4	F
GFN2-xTB [136]	F	PBE0-2 [137]	F	VV10 [138]	F
HCTH/120 [139]	A	PBE0-D2 [61]	B	ωB97 [140]	F
HCTH/120-D3(0)	A	PBE0-D3(0)	B	ωB97M-V [141]	F
HCTH/120-D3(BJ)	A	PBE0-D3(BJ)	B	ωB97M(2) [142]	F
HCTH/147 [139]	A	PBE0-D3(CSO)	B	ωB97X [140]	F
HCTH/407 [143]	A	PBE0-D3M(BJ)	C	ωB97X-D [144]	F
HCTH/93 [80]	A	PBE0-D4	B	ωB97X-D3 [145]	F
HF [48,49,50]	F	PBEh-3c [146]	F	ωB97X-V [147]	F
HF-3c [148]	F	PBEOP [57,120]	F	ωM05-D [149]	F
HF-D3(0)	F	PBEsol [94]	F	ωM06-D3 [145]	F
HF-D3(BJ)	F	PBEsol-D3(0)	F	X3LYP [150]	B
HF-NL [68]	F	PBEsol-D3(BJ)	F	XYG3 [151]/XYGJ-OS [152]	F

**Table 2 molecules-28-03487-t002:** General performance of selected functional approximations for the Por21 database for 25 of the “most used, most suggested” (MUMS) functionals in the literature. All values are mean unsigned errors (MUEs) in kcal/mol calculated from the CASPT2 reference energies.

MUMS Functional:	Type *^a^*	Por21	PorSS11	PorBE10
r^2^SCANh	GH-mGGA	10.8	7.49	14.4
M06-L	mGGA	11.8	11.9	11.6
MN15-L	mGGA	11.9	17.9	5.26
r^2^SCAN	mGGA	13.4	13.1	13.6
M06	GH-mGGA	15.1	17.9	12.0
PBE0	GH-GGA	16.1	17.1	15.0
r^2^SCAN0	GH-mGGA	17.3	17.4	17.1
B3LYP	GH-GGA	19.1	21.1	16.8
B97M-V	mGGA	19.8	20.6	18.9
PW6B95	GH-mGGA	22.2	20.4	24.2
SCAN	mGGA	22.6	21.3	24.1
TPSSh	GH-mGGA	22.9	26.7	18.8
BLYP	GGA	25.6	27.6	23.4
PBE	GGA	26.3	27.8	24.6
TPSS	mGGA	26.7	30.5	22.5
MN15	GH-mGGA	26.8	30.5	22.7
B2PLYP	DH	27.2	21.3	33.6
BP86	GGA	27.5	29.3	25.5
CAM-B3LYP	RSH-GGA	28.5	32.5	24.2
ωB97M-V	RSH-mGGA	31.5	36.2	26.4
M06-2X	GH-mGGA	31.8	36.1	27.2
ωB97X-V	RSH-GGA	36.0	35.4	36.7
PWPB95-D4	DH	37.1	31.5	43.3
ωB97M(2)	DH	42.0	46.7	36.7
B3LYP*	GH-GGA	43.9	20.8	69.4

*^a^* GGA: generalized gradient approximation; mGGA: meta-GGA; GH: global hybrid; RSH: range-separated hybrid; DH: double-hybrid.

**Table 3 molecules-28-03487-t003:** Spin state energy differences for the FeP system with 25 of the “most used, most suggested” (MUMS) functionals in the literature. All values are in kcal/mol. WFT = Wavefunction theory.

MUMS Functional:	Type *^a^*	FeP^5^ → FeP^3^	FeP^5^ → FeP^1^
r^2^SCANh	GH-mGGA	−1.76	34.9
M06-L	mGGA	−1.35	25.6
MN15-L	mGGA	12.6	48.7
r^2^SCAN	mGGA	−9.58	26.3
M06	GH-mGGA	−4.49	27.2
PBE0	GH-GGA	−8.69	27.2
r^2^SCAN0	GH-mGGA	−3.83	33.2
B3LYP	GH-GGA	−14.6	18.6
B97M-V	mGGA	−21.5	10.8
PW6B95	GH-mGGA	−12.4	19.6
SCAN	mGGA	−24.0	5.44
TPSSh	GH-mGGA	−19.7	13.8
BLYP	GGA	−16.4	17.4
PBE	GGA	−15.9	19.1
TPSS	mGGA	−26.2	6.68
MN15	GH-mGGA	−45.2	−9.77
B2PLYP	DH	−19.7	14.3
BP86	GGA	−13.7	19.6
CAM-B3LYP	RSH-GGA	−11.4	24.2
ωB97M-V	RSH-mGGA	−57.5	106.
M06-2X	GH-mGGA	−49.4	−11.8
ωB97X-V	RSH-GGA	−53.9	108.
PWPB95-D3(BJ)	DH	−62.1	93.0
ωB97M(2)	DH	−80.9	−54.0
B3LYP*	GH-GGA	−14.6	18.6
CCSD(T)	WFT	2.30	33.0
CASPT2	WFT, Reference	7.00	39.9

*^a^* GGA: generalized gradient approximation; mGGA: meta-GGA; GH: global hybrid; RSH: range-separated hybrid; DH: double-hybrid.

## Data Availability

All the data are available within the manuscript and the Appendix A.

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
