# Peer review of "Comparison of the Performance of Density Functional Methods for the Description of Spin States and Binding Energies of Porphyrins"

_molecules, 2023, doi:10.3390/molecules28083487_

Round 1
Reviewer 1 Report
The paper by Morgante and Peverati provides an in-depth analysis of the accuracy of the DFT methods to predict the spin-state and binding energies in Fe, Co and Mn-metalloporphyrines. The results are sound, instructive and provide a solid base for the future studies.
Still, I have some problem with the way the results are presented. I think it may be considered to difficult to read for a wider community. I think there are few corrections in this respect that may help.
1. Make it clear in any table and in the Abstract that reference are the reference energies were obtained with the CASPT2.
2. Please state all 11 spin states that are considered for Mn(II), Co(II), Fe(II) and Fe(III) complexes. Please give one example for each.
2a. Please give one example of binding energies.
3. The authors write (lines 85-86) "To analyze the results in more detail, the spin states data have been grouped in a subset 85 called PorSS11, while the bond energies data were grouped in the PorBE10 subset". However, the key Table 1 shows the results obtained for the basic set Por21, leaving an ueasy feeling that the spin states energies are compared with the binding energies. The performance for both is discussed to some extent below the Table. Wouldn't that make things more clear if the grades in Table 1 were given for spin state energies and binding energies separately? If the grades match perfectly for both, please state it explicitly in Table caption.
I hope the above correction may help the reader.
Author Response
Reviewer 1
The paper by Morgante and Peverati provides an in-depth analysis of the accuracy of the DFT methods to predict the spin-state and binding energies in Fe, Co and Mn-metalloporphyrines. The results are sound, instructive and provide a solid base for the future studies.
We thank the reviewer for their comment.
Still, I have some problem with the way the results are presented. I think it may be considered to difficult to read for a wider community. I think there are few corrections in this respect that may help.
- Make it clear in any table and in the Abstract that reference are the reference energies were obtained with the CASPT2.
The CASPT2 method is now mentioned in the Abstract. The new sentence reads: "The assessment employs the Por21 database of high-level computational data (CASPT2 reference energies taken from the literature)". We also added specific reference to the CASPT2 reference energies to the caption of Tables 1 and 2.
- Please state all 11 spin states that are considered for Mn(II), Co(II), Fe(II) and Fe(III) complexes. Please give one example for each.
2a. Please give one example of binding energies.
We added a sentence in the “Materials and Methods” section stating which spin states are considered. The new sentence reads: "The spin multiplicities considered are sextuplet, quadruplet, and doublet for Mn(II) and Fe(III), quartet and doublet for Co(II), and quintet, triplet, and singlet for Fe(II)." We also added a chemical equation for the binding energies.
- The authors write (lines 85-86) "To analyze the results in more detail, the spin states data have been grouped in a subset 85 called PorSS11, while the bond energies data were grouped in the PorBE10 subset". However, the key Table 1 shows the results obtained for the basic set Por21, leaving an uneasy feeling that the spin states energies are compared with the binding energies. The performance for both is discussed to some extent below the Table. Wouldn't that make things more clear if the grades in Table 1 were given for spin state energies and binding energies separately? If the grades match perfectly for both, please state it explicitly in Table caption.
We thank the reviewer for this helpful remark. We added two new tables in the supplementary material that give grades for the PorSS11 and PorBE10 subsets separately. The following statement was added to the discussion: "We note in passing that most of the grades obtained for the entire Por21 database are transferable to the PorSS11 and PorBE10 datasets (the individual rankings for the subsets are reported in Tables S1 and S2 in the supplementary data).” while the caption of Table 1 now reads: "See also Tables S1 and S2 in the supplementary data for results on the PorSS11 and PorBE10 subsets.". We decided to present the separate subsets in the supplementary data to avoid making the main text too long.

Reviewer 2 Report
In this manuscript, the authors are studying how 240 computational approaches (mainly density functionals) can reproduce the spin states and binding energies of transition metal porphyrins. The complexes are taken from the Por21 database (gathered by the authors): energies are computed at the CASPT2 level, while geometries are optimized using PBE0 for FeP or BP86 for other molecules. The conclusions are quite saddening as no method is able to reach an MUE less than 5 kcal/mol.
This work gathers lots of computations and is thus impressive. However, I would like the authors to address the following major points:
- - the authors use the CASPT2 values from Pierloot et al. as reference energies. However, in their JCTC 2017, Pierloot et al. used CCSD(T) values as reference: why not using CCSD(T) values also?
- - out of 11 spin states, 3 concern the FeP spin states for which CASPT2 calculations give the quintet state (5A1g ) as the ground state. This is still debated as recent articles predict a triplet state to be the ground state, such as: O. Weser et al. J. Chem. Theory Comput. 2022, 18, 1, 251–272 or D. Truhlar et al. J. Phys. Chem. A 2022, 126, 24, 3957–3963. P. Hobza et al. (Phys. Chem. Chem. Phys., 2020, 22, 17033-17037) have shown that the predicted ground state depends on the Fe-N bond distance: the quintet state is the most stable for long Fe-N distance, while the triplet state is the most stable at short Fe-N distance.
The authors might want to be more cautious about the performances of the tested methods for those spin states? Or may be check them for different Fe-N distances?
- Also for FeP: why is the 3A2g state not considered here?
Some more minor points:
It is a bit strange that the authors never give the computed relative energies of the spin state or the actual binding energies: this should be given in the Supporting Information together with the signed and unsigned errors.
Also, the provided Excel file is quite difficult to use as only the ‘codename’ of the molecules are given, and not their actual name: Por21_4 is in fact “-1,PorSS11_FeP_5A1g,1,PorSS11_FeP_3Eg” which shall be interpreted as the energy of the 3Eg state relative to the 5A1g state. But to know this, you have to look at the POR21 database on the authors github. I suggest that the “simple” definition could be added in the Excel file, together with the computed spin state relative energies.
Author Response
Reviewer 2
In this manuscript, the authors are studying how 240 computational approaches (mainly density functionals) can reproduce the spin states and binding energies of transition metal porphyrins. The complexes are taken from the Por21 database (gathered by the authors): energies are computed at the CASPT2 level, while geometries are optimized using PBE0 for FeP or BP86 for other molecules. The conclusions are quite saddening as no method is able to reach an MUE less than 5 kcal/mol.
We thank the reviewer for their time and care in reviewing our manuscript.
This work gathers lots of computations and is thus impressive. However, I would like the authors to address the following major points:
1 the authors use the CASPT2 values from Pierloot et al. as reference energies. However, in their JCTC 2017, Pierloot et al. used CCSD(T) values as reference: why not using CCSD(T) values also?
This is a good point. In fact, we asked ourselves the same question when putting together the database a few years ago. Ultimately, we decided to adopt the CASPT2 reference data for consistency with the PorBE10 subset and because most of the states involved potential multireference character. Indeed, we agree with the referee on the fact that our potential readers would benefit for a more comprehensive discussion on this topic. For this reason, we added a new paragraph discussing the results for FeP and the reliability of the reference data (see also our answer to point #2 and #3 below).
2 out of 11 spin states, 3 concern the FeP spin states for which CASPT2 calculations give the quintet state (5A1g ) as the ground state. This is still debated as recent articles predict a triplet state to be the ground state, such as: O. Weser et al. J. Chem. Theory Comput. 2022, 18, 1, 251–272 or D. Truhlar et al. J. Phys. Chem. A 2022, 126, 24, 3957–3963. P. Hobza et al. (Phys. Chem. Chem. Phys., 2020, 22, 17033-17037) have shown that the predicted ground state depends on the Fe-N bond distance: the quintet state is the most stable for long Fe-N distance, while the triplet state is the most stable at short Fe-N distance.
The authors might want to be more cautious about the performances of the tested methods for those spin states? Or may be check them for different Fe-N distances?
3 Also for FeP: why is the 3A2g state not considered here?
We agree with the reviewer that these issues are indeed important to discuss. For this reason, we added a section titled "Discussion on reference energies and chemical accuracy for transition metals" to assess this issue. We specifically included in the new section a table where we compare the spin state energy differences of iron porphyrin (together with the CCSD(T) results mentioned above) and we added a comment explaining which states are included in Por21 and why.
Some more minor points:
4 It is a bit strange that the authors never give the computed relative energies of the spin state or the actual binding energies: this should be given in the Supporting Information together with the signed and unsigned errors.
We agree with the reviewer, this was a mistake in our original Supporting Information that we have now fixed by adding two new sheets in the Excel file that list the calculated values and the signed errors for each method.
5 Also, the provided Excel file is quite difficult to use as only the ‘codename’ of the molecules are given, and not their actual name: Por21_4 is in fact “-1,PorSS11_FeP_5A1g,1,PorSS11_FeP_3Eg” which shall be interpreted as the energy of the 3Eg state relative to the 5A1g state. But to know this, you have to look at the POR21 database on the authors github. I suggest that the “simple” definition could be added in the Excel file, together with the computed spin state relative energies.
The reviewer is right on this point as well. We updated the Excel file to show which chemical process is considered in each datapoint. We also updated the name of each datapoint to reflect the labeling used in the Por21 database (Por21_XX) and in each subset (PorSS11_XX or PorBE10_XX). For example, the labels given for Por21_4 are "Por21_4, PorSS11_4, FeP5 ⟶ FeP3". Similar labels were adopted for each datapoint.

Reviewer 3 Report
This article is an important and extensive investigation devoted to quantum chemical calculations of porphyrin complexes of some d-metals. Indeed, the selection of a suitable calculation method is often a time-consuming task for a chemist, especially when it comes to high-precision calculations. Of course, this research deserves a lot of attention and is of high value both from the point of view of the extensive range of methods used, and from the point of view of the revealed limitations in the accuracy of calculations.
Author Response
Reviewer 3
This article is an important and extensive investigation devoted to quantum chemical calculations of porphyrin complexes of some d-metals. Indeed, the selection of a suitable calculation method is often a time-consuming task for a chemist, especially when it comes to high-precision calculations. Of course, this research deserves a lot of attention and is of high value both from the point of view of the extensive range of methods used, and from the point of view of the revealed limitations in the accuracy of calculations.
We thank the reviewer for their time and care in reviewing our manuscript.

Reviewer 4 Report
The present work verified the performance of 240 exchange-correlation (XC) functional in the study of spin states and binding properties of iron, manganese, and cobalt porphyrins. This study is interesting and has merit for publication in Molecules. However, it is suggested that:
(1) In the introduction, the authors should describe in general terms why these large varieties of XC functionals were developed so far.
(2) Also, describe the motivation for the choice of the XC functionals employed in the present work.
(3) Give more details of the electronic structure method employed and the motivation for the choice of such quantity.
(4) Try to be clear about what the most influences in analyzed results are: XC functional, electronic structure method, or both.
(5) Another fact that is not clear is if the molecular geometry was optimized with all different XC functionals. Describe this better in the manuscript.
(6) Give the appropriate reference for all XC functional describe in Table 1.
Author Response
Reviewer 4
The present work verified the performance of 240 exchange-correlation (XC) functional in the study of spin states and binding properties of iron, manganese, and cobalt porphyrins. This study is interesting and has merit for publication in Molecules.
However, it is suggested that:
(1) In the introduction, the authors should describe in general terms why these large varieties of XC functionals were developed so far.
(2) Also, describe the motivation for the choice of the XC functionals employed in the present work.
(3) Give more details of the electronic structure method employed and the motivation for the choice of such quantity.
To address all three of these points, we dug through the parameters of all computational software available to us and we were able to include an additional 10 functionals, bringing the total to an unprecedented 250 methods tested on metalloporphyrins. At this point, we believe that there is no need to motivate the choice of XC functional employed in the present work, just simply because there was no choice: We tested every single method that is available for calculations of electronic structure properties of metalloporphyrins. The motivation of such a high number of methods is clear when considering that this wants to be a comprehensive benchmark study. We also feel that it is well beyond the scope of this work to explain the development philosophy behind all 250 different methods, since this is not a review article. We are confident, however, that every reader will find their favorite methods among the list of our results, and they will be able to bridge the knowledge gap using the primary sources quoted in our article. We have added some discussion to the ‘Materials and Methods’ section at the end of the manuscript to clarify these points.
(4) Try to be clear about what the most influences in analyzed results are: XC functional, electronic structure method, or both.
We believe the new section titled “Discussion on reference energies and chemical accuracy for transition metals” that we added in response to the remarks of Reviewer #2 addresses this point as well.
(5) Another fact that is not clear is if the molecular geometry was optimized with all different XC functionals. Describe this better in the manuscript.
We apologize if this was not clear in the previous version of the manuscript. We now explicitly state that "The molecular structures are taken from the original publications, and they were not re-optimized in this work."
(6) Give the appropriate reference for all XC functional describe in Table 1.
We are puzzled by this comment, since all the references for all methods were indeed listed in Table 1 in our original submission. We made sure to keep them in our revised manuscript as well, and we will work with the editorial team to make sure that no mistake happens when the article is exported and sent to the reviewers and ultimately accepted for publication.

Round 2
Reviewer 2 Report
In this new version of the manuscript, the authors have answered all my comments. I am thus happy to recommend publication of this nice work in its present form.